# SafeDec: Constrained Decoding for Safe Autoregressive Generalist Robot Navigation Policies

**Parv Kapoor** [1] **Akila Ganlath** [2] **Michael Clifford** [2] **Changliu Liu** [1] **Sebastian Scherer** [1] **Eunsuk Kang** [1]

## Abstract

Recent advances in end-to-end, multi-task robot policies based on transformer models have demonstrated impressive generalization to real-world embodied navigation tasks. Trained on vast datasets of simulated and real-world trajectories, these policies map multimodal observations directly to action sequences for physical execution. Despite promising real-world capabilities, these models are still data-driven and, therefore, lack explicit notions of behavioral correctness. We address this gap by introducing **SafeDec**, a constrained decoding framework for autoregressive, transformer-based robot navigation policies that enforces safety specifications expressed as Signal Temporal Logic (STL) formulas. Our method ensures that generated actions provably satisfy STL specifications under assumed dynamics at runtime without retraining while remaining agnostic of the underlying policy. We evaluate **SafeDec** on tasks from the CHORES benchmark for state-of-the-art embodied navigation policies across hundreds of procedurally generated environments and show that our decoding-time interventions are useful not only for filtering unsafe actions but also for conditional action generation. Videos are available at **constrained-robot-fms.github.io**.

## 1. Introduction

Recent advances in developing large transformer-based models for embodied navigation have enabled general-purpose policies that map multi-modal inputs such as RGB images, natural language instructions, and proprioceptive inputs to action sequences (Hu et al., 2023). For example, Shortest Path Oracle Clone (SPOC) (Ehsani et al., 2024), PoliFormer (Zeng et al., 2025), Flare (Hu et al., 2025), and LEO (Huang et al., 2024) exhibit impressive zero-shot generalization for embodied navigation tasks such as open vocabulary object-goal navigation ("find a mug") and attribute-conditioned navigation ("locate the chair closest to the refrigerator in the kitchen"). These models have also shown promising transfer for robot navigation in real-world settings (Zeng et al., 2025; Huang et al., 2024). However, these policies are primarily data-driven and lack any explicit notion of safety. Although policies may implicitly exhibit safety-related behaviors depending on patterns in their training data, there is no formal guarantee that they will consistently behave safely in all situations. This limitation hinders the deployment of such policies in the physical world where adherence to explicit safety rules and regulatory constraints is essential.

Formal specifications have long been used to specify contextual safety requirements for robotic deployments (Menghi et al., 2019; Farrell et al., 2018). Specifically, *temporal logics* (*TL*) (Pnueli, 1977) provide a structured language for specifying rich safety constraints on robot behavior, ranging from simple invariants (e.g., remaining within permitted operational zones or avoiding dangerous obstacles) to contextual and history-dependent specifications. For example, a robot may be required to avoid certain regions until visiting another region (temporal ordering), permanently avoid a region after entering a restricted area (conditional safety) or never return to a region once it has exited. Such specifications naturally capture real-world operational rules that depend not only on the robot's current state, but also on what situations the robot has previously encountered.

Although TL has seen success in classical robotic planning and reinforcement learning for safety constraint satisfaction, its use for enforcing safety for large transformer-based robot policies remains limited (Manganaris et al., 2026). Additionally, retraining or fine-tuning these large pre-trained policies to directly embed such non-Markovian TL safety specifications is challenging (Kapoor et al., 2024). First, retraining policies is a costly endeavor in terms of computational resources and data requirements. Moreover, due to the stochastic nature of these models, it is difficult to guarantee strict satisfaction of safety constraints through training alone. Hence, there is a pressing need for methods

---

[1]Carnegie Mellon University, Pittsburgh, Pennsylvania, USA. [2]Toyota InfoTech Labs, Mountain View, California, USA. Correspondence to: Parv Kapoor <parvk@cs.cmu.edu>.

*Proceedings of the 43rd International Conference on Machine Learning*, Seoul, South Korea. PMLR 306, 2026. Copyright 2026 by the author(s).

that can reliably enforce safety specifications at inference time without policy retraining.

In the field of natural language processing, syntactic constraints have been successfully enforced by applying *constrained decoding* at inference time (Willard & Louf, 2023; Beurer-Kellner et al., 2023; AI, 2023). These approaches typically mask out tokens that violate a syntactic constraint defined over token sequences. For example, regular expressions (regex) represent a widely used form of syntactic constraint, requiring that generated token sequences conform to predefined structural patterns (Willard & Louf, 2023; Beurer-Kellner et al., 2023). Inspired by this line of work, we extend the paradigm of constrained decoding to enforce contextual safety constraints over action trajectories in dynamical systems. We propose *safety specification aligned decoding (SafeDec)* for transformer-based policies that ensures generated action sequences provably satisfy Signal Temporal Logic (STL) (Maler & Nickovic, 2004) specifications under assumed dynamics. Our key insight is that decoding-time interventions can be used not just to filter unsafe actions, but to *condition the generation process itself* on specification satisfaction. This conditioning is critical because it steers the model toward generating safety specification satisfying actions rather than relying on post hoc rejection. To enforce safety specifications, we leverage the formal semantics of STL to evaluate candidate actions at runtime and mask those that lead to future violations. Our method is agnostic to the underlying foundation model, requiring only two properties: (1) access to the decoding-layer logits during inference, and (2) access to an approximate dynamics model to predict future states. In this work, we focus on safety enforcement for embodied navigation policies in indoor environments. We also focus on STL *safety* (Alpern & Schneider, 1987) specifications without liveness operators, since such specifications are sufficient to model a wide range of safety constraints encountered in robot learning (He et al., 2024; Yun et al., 2025; Kapoor et al., 2025b; Zhao et al., 2024). To the best of our knowledge, this is the first work to enforce STL safety specifications on transformer-based navigation policies at inference time using constrained decoding.

## 2. Preliminaries

### 2.1. Signal Temporal Logic

Signal Temporal Logic (STL) is an expressive framework for defining properties and reasoning over continuous-time real-valued (Maler & Nickovic, 2004). Formally, $(\mathbf{s}, t) \models \phi$ denotes that a signal $\mathbf{s}$ satisfies the STL formula $\phi$ at time $t$. An atomic predicate of an STL formula is represented by inequalities of the form $\mu(\mathbf{s}(t)) > 0$. The truth value of the predicate $\mu$ is equivalent to $\mu(\mathbf{s}(t)) > 0$. Note that with slight abuse of notation, $\mu$ represents both the predicate and

a function of the trajectory $\mathbf{s}(t)$. Any STL formula consists of Boolean and temporal operations on predicates, and the syntax of STL formulas is defined recursively as follows:

$$\phi := \mu \mid \neg\mu \mid \phi \wedge \psi \mid \phi \vee \psi \mid \mathbf{G}_{[a,b]}\,\psi \mid \mathbf{F}_{[a,b]}\,\psi \mid \phi\,\mathbf{U}_{[a,b]}\,\psi$$

where $\psi$ and $\phi$ are STL formulae, $\mathbf{G}$ denotes the globally operator, $\mathbf{F}$ the eventually operator, and $\mathbf{U}$ is the until operator. For example, $\mathbf{s} \models \mathbf{G}_{[a,b]}\psi$ specifies that $\psi$ must be in all times in the given interval, $t \in [a, b]$ of the signal $\mathbf{s}$. Similarly, the operator *until* in $\mathbf{s} \models \phi\mathbf{U}_{[a,b]}\psi$ defines that $\phi$ must be true until $\psi$ becomes true within a time interval $[a, b]$.

Given a signal $s_t$ representing a signal starting at time t, the Boolean semantics of satisfaction of $s_t \models \phi$ are defined inductively as follows:

$$
\begin{aligned}
s_t \models \mu &\iff \mu(s(t)) > 0 \\
s_t \models \neg\varphi &\iff \neg(s_t \models \varphi) \\
s_t \models \varphi_1 \wedge \varphi_2 &\iff (s_t \models \varphi_1) \wedge (s_t \models \varphi_2) \\
s_t \models \mathbf{F}_{[a,b]}(\varphi) &\iff \exists t' \in [t+a, t+b] \text{ s.t. } s_{t'} \models \varphi \\
s_t \models \mathbf{G}_{[a,b]}(\varphi) &\iff \forall t' \in [t+a, t+b] \text{ s.t. } s_{t'} \models \varphi
\end{aligned}
$$

Apart from Boolean semantics, quantitative semantics are defined for a signal to compute a real-valued metric called *robustness*, i.e., the strength of satisfaction or violation.

Given a signal $s_t$ representing a signal starting at time t, the quantitative semantics of satisfaction of $s_t \models \phi$ are defined inductively as follows:

$$
\begin{aligned}
\rho(s_t, \mu_c) &= \mu(x_t) - c \\
\rho(s_t, \neg\varphi) &= -\rho(s_t, \varphi) \\
\rho(s_t, \varphi_1 \wedge \varphi_2) &= \min(\rho(s_t, \varphi_1), \rho(s_t, \varphi_2)) \\
\rho(s_t, \mathbf{F}_{[a,b]}(\varphi)) &= \max_{t' \in [t+a, t+b]} \rho(s_{t'}, \varphi) \\
\rho(s_t, \mathbf{G}_{[a,b]}(\varphi)) &= \min_{t' \in [t+a, t+b]} \rho(s_{t'}, \varphi)
\end{aligned}
$$

### 2.2. Constrained Decoding in Transformers

A large variety of autoregressive transformer-based models generate final outputs by producing a probability distribution over the model vocabulary at each timestep. Then, through the process of *decoding*, tokens are selected to maximize the overall likelihood of an output sequence. In standard decoding, this maximization can be performed by either greedily selecting the most probable token at each step or by using a beam search to maintain multiple high-likelihood candidates. However, this often leads to degenerate output sequences that are repetitive (Holtzman et al., 2019). A common approach is to use sampling strategies like top $k$ (Fan et al., 2018), and nucleus sampling (Holtzman et al., 2019) that introduce stochasticity to encourage more diverse

outputs. Constrained decoding (Hokamp & Liu, 2017) modifies this probabilistic selection by pruning invalid tokens to ensure that the generated sequences satisfy predefined constraints. These constraints are often syntactic, such as regular expressions, JSON formatting, or programming language grammars (Welleck et al., 2024). There is also recent work on enforcing *semantic* constraints that ensure coherence of the output or alignment with specific knowledge bases (Peyrard et al., 2024). Formally, constrained decoding can be seen as maximizing the probability of the output sequence subject to a constraint $\mathcal{C}$: $\arg\max_{y \in \mathcal{Y}_\mathcal{C}} P(y \mid x)$ where $\mathcal{Y}_\mathcal{C}$ is the set of sequences satisfying $\mathcal{C}$.

## 3. Specification-Guided Constrained Decoding

In this section, we introduce a novel problem formulation for SafeDec in autoregressive transformer-based policies. First, we highlight the challenge in specification checking for these policies in contrast to traditional syntactical constraint checking adopted by LLMs, and our solution to remedy it. Then, we propose two novel inference-time techniques for specification aligned decoding: *Hard Constrained Decoding* (**HCD**) and *Robustness Constrained Decoding* (**RCD**).

### 3.1. Problem Statement

As highlighted in the background section, existing techniques in constrained decoding for language models enforce *syntactic constraints* defined over tokens, such as conforming to a context-free grammar or matching a regular expression. In these setups, constraint checking can be performed in the model's token space.

In contrast, transformer-based policies operate in a physical environment and constraints (captured via temporal logic) are defined over state variables. Since a large class of end-to-end policies solely propose action sequences, specification checking can only be performed as actions are executed and the environment is simulated forward. In this case, constraint checking cannot be done solely in the token space and requires environmental feedback or a dynamics stepping function. Hence, we leverage an approximate first order dynamics function to compute specification satisfaction of different action sequences proposed by these policies.

Consider a discrete dynamical system with states $x_t \in \mathbb{R}^n$ and actions $a_t \in \mathbb{A}$ at time step $t$. The system's dynamics are defined by $x_{t+1} = f(x_t, a_t)$ where $f : \mathbb{R}^n \times \mathbb{A} \to \mathbb{R}^n$ maps the current state $(x_t \in \mathbb{R}^n)$ and a discrete action $(a_t \in A)$ to the next state $x_{t+1} \in \mathbb{R}^n$. This system is controlled by a policy that selects an action $a_t$ at each time step based on observations and task context such as user-provided natural-language instructions or goal waypoints.

In this work, we focus on robot policies that generate actions

based on multi-modal inputs, including sensor observations (e.g., RGB, depth, LiDAR) and natural language instructions. Let $\mathcal{I}_t$ represent the aggregated input at timestep $t$. Given the history of encoded inputs up to time $t$, a transformer-based policy parameterized by $\theta$ predicts embeddings for the next $T-t$ actions:

$$\{\hat{e}_{a_{t+k}}\}_{k=1}^{T-t} = \text{Transformer}_\theta\big(\{e_{\mathcal{I}_\tau}\}_{\tau=0}^t\big).$$

Each predicted action embedding is decoded into an action $\hat{a}_{t+k} \in \mathbb{A}$, resulting in a predicted action sequence $\{\hat{a}_{t+1}, \ldots, \hat{a}_T\}$, where $T$ denotes the planning horizon.

Now, consider that the system is required to satisfy requirements encoded using an STL formula $\varphi$ defined over the state variables of the system. Formally, the goal is to ensure that the resulting trajectory satisfies the specification $\varphi$:

$$\{(x_0, \hat{a}_0), \ldots, (x_T, \hat{a}_T)\} \models \varphi$$

Most of the existing techniques for specification enforcement perform post-hoc manipulation of proposed actions (after sampling from the policy) by filtering or rejecting action sequences that violate the specification $\varphi$. Although manipulation after sampling can ensure specification satisfaction, it can favor safe but low-likelihood actions and unintentionally suppress feasible safe behaviors that the model would have selected if safety constraints had been considered pre-sampling. This undermines the inductive biases learned during pretraining and lead to degenerate and brittle behaviors. A similar problem was highlighted when ensuring compliance with logical constraints for large language models (LLM) in Park et al. (2024). Additionally, these models decode actions sequentially, where each action $a_t$ is conditioned on previously generated tokens $a_{<t}$. Post hoc manipulation can disrupt this causal chain and lead to a mismatch between the model's internal hidden state and the executed sequence. Hence, we propose the following problem statement:

> *How can we enforce temporal logic constraints during action generation in robot foundation models such that the output sequence (1) satisfies an STL specification $\varphi$, and (2) remains faithful to the model's autoregressive distribution $\pi(a_{1:T} \mid \mathcal{I}_{1:T})$?*

Let $\pi(a_{0:T})$ be the *unconstrained* action-sequence distribution produced by the transformer-based policy's decoder (e.g. the softmax over logits generated by the Transformer). We define the ideal constrained distribution over action sequences as:

$$Q_{\pi,\varphi}(a_{0:T}) = \frac{\pi(a_{0:T}) \cdot \mathbf{1}[(x_{1:T}, a_{0:T}) \models \varphi]}{\sum_{a'_{0:T}} \pi(a'_{0:T}) \cdot \mathbf{1}[(x'_{1:T}, a'_{0:T}) \models \varphi]} \quad (1)$$

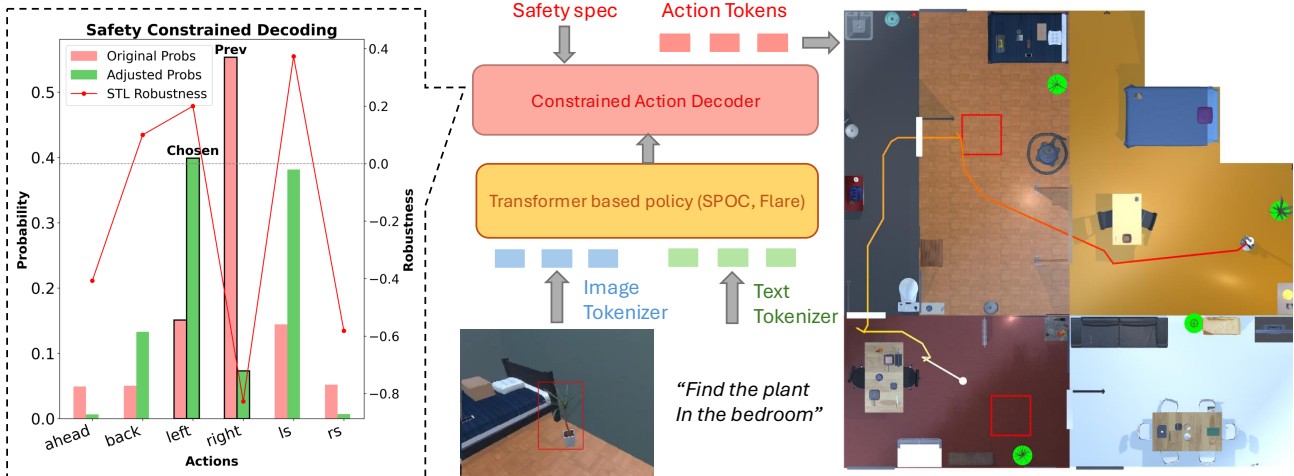

*Figure 1.* **Overview of our specification aligned decoding framework.** Given multimodal observations (RGB images and language goals), a pretrained transformer-based navigation policy generates candidate actions. Our constrained decoder then filters or reweights these actions using robustness scores from a user-defined STL safety specification. **Left:** Original action probabilities (red) from the pretrained policy are modified by SafeDec using STL robustness: actions predicted to violate the safety specification receive reduced weights, while safer actions are boosted. The chosen safe action and adjusted probabilities (green) are highlighted. **Right:** An example navigation episode where the robot starts in the kitchen (white dot) and moves toward the bedroom to locate the target object (green marker middle-right), while avoiding user-defined hazardous regions (red zones) throughout the trajectory.

where $x_{1:T}$ denotes the state trajectory induced by the system dynamics under actions $a_{0:T}$ and $\mathbf{1}[\cdot]$ is the indicator function that returns 1 iff the trajectory-action pair satisfies the specification. Equation 1 is the exact Bayesian conditioning of $\pi$ on the event that the generated rollout satisfies $\varphi$. Hence, sampling from $Q_{\pi,\varphi}$ would give sequences that (i) inherit the original model's preferences encoded in $\pi$ and (ii) *guarantee* specification satisfaction.

In this work, we propose a technique to overcome the drawbacks of post-hoc safety enforcement methods (such as filtering) by leveraging constrained decoding techniques. Specifically, we propose `SafeDec` : A constrained decoding strategy that integrates STL specifications into the policy's action selection process, ensuring satisfaction while remaining as close as possible to the base model distribution.

### 3.2. Hard Constrained Decoding

As highlighted in the background section, in the final layer, predictions are detokenized and a projection layer converts the embeddings into logits over the vocabulary space. These logits are further converted into a probability distribution using a softmax operation. In prior work, for structured output generation in LLMs, some invalid tokens are masked based on syntactical constraints or other criteria (Welleck et al., 2024; Park et al., 2024). This is done by setting their logit value as $-\infty$ before the softmax operation is applied. For HCD, we use a similar approach as constrained decoding literature (Welleck et al., 2024) and mask out predicted

action tokens that violate our given STL specification $\varphi$ during sequential generation. Formally, to enforce the STL specification $\varphi$ during sequential generation, we adjust the logits at each timestep $t + k$ as follows:

Let $\mathbf{z}_{t+k}$ denote the logits at timestep $t + k$. For each action choice $i$ at timestep $t + k$, we define:

$$z_{t+k}^{(i)} = \begin{cases} -\infty, & \text{if } \hat{\tau}_{0:t+k}^{(i)} = (x_0, x_1, .., \hat{x}_{t+k}^{(i)}) \text{ violates } \varphi \\ z_{t+k}^{(i)}, & \text{otherwise} \end{cases}$$

where $\hat{x}_{t+k}^{(i)} = f(\hat{x}_{t+k-1}, \hat{a}_{t+k}^{(i)})$.

Here, $t$ is the current decision step, $k$ is an index for the look-ahead step $t + k$ within a planning horizon of length $T$ ($k \in [1..T]$), $\hat{a}_{t+k}^{(i)}$ is the action mapping to the token i and $\hat{x}_{t+k}^{(i)}$ is the next state value upon taking this action. This next state is elicited using a simple dynamics model ($f$) as highlighted in the previous section. Adjusting logits in this fashion ensures that any invalid token with respect to the safety specification will have zero probability of being selected after applying the softmax function.

### 3.3. Robustness Constrained Decoding

HCD ensures compliance but can lead to compromising task success, which can be undesirable. A similar trade-off was observed by Liu et al. (2021) when probability space-steering preserved model fluency while reducing toxic continuations compared with hard-filtering strategies that

*Table 1.* STL specifications used in our evaluation. Each region $R_i$ denotes a spatial predicate over $(x, z)$.

| Spec name | STL formula | Description |
|---|---|---|
| $\phi_{\text{avoid}}$ | $\mathbf{G}\left(\bigwedge_{i=1}^{N} \neg R_i\right)$ | Avoidance: the robot must always stay outside all unsafe regions. |
| $\phi_{\text{ordered}}$ | $\mathbf{G}(\neg R_2) \vee (\neg R_2 \ \mathbf{U} \ R_1)$ | Weak-until ordering: the robot must avoid region $R_2$ until reaching region $R_1$, or avoid $R_2$ forever. |
| $\phi_{\text{no\_return}}$ | $\mathbf{G}(R_0 \Rightarrow \mathbf{G}(\neg R_0 \Rightarrow \mathbf{G}\neg R_0))$ | Conditional no-return: if the robot is in region $R_0$, then once it leaves $R_0$, it may never re-enter. |
| $\phi_{\text{geofence}}$ | $\mathbf{G}\left(\bigvee_{i=1}^{N} R_i\right)$ | Geofencing: the robot must always remain within at least one allowed region. |

inflated perplexity and eroded diversity. Hence, we propose an alternative approach, called RCD, where we leverage the quantitative semantics of STL specifications (robustness). Unlike HCD, which applies hard masking to completely remove unsafe actions, RCD softly guides the model toward safer actions by incorporating robustness scores that reflect the degree of satisfaction of $\varphi$. This is similar to the approach proposed in Liu et al. (2021) where the next-token distributions were re-weighted based on the utility scores provided by another language model. Our utility scores are quantified by the robustness function( $\rho(\langle x_0, x_1, \ldots, x_t \rangle, \varphi)$ ) that returns a real-valued score indicating how well a predicted state satisfies the specification. Positive robustness values denote specification satisfaction, while negative values capture the degree of violation. A formal definition of robustness in line with the STL quantitative semantics is provided in Section 2.

First, we compute a robustness score for each candidate action: $r_{t+k}^{(i)} = \rho(\langle x_0, x_1, \ldots, x_{t+k-1}, \hat{x}_{t+k}^{(i)} \rangle, \varphi)$ where $\rho(\cdot, \varphi)$ is the STL robustness metric, and $\hat{x}_{t+k}^{(i)}$ is the predicted next state under action $\hat{a}_{t+k}^{(i)}$. This robustness score $r_{t+k}^{(i)}$ quantifies how well each candidate action satisfies the specification $\varphi$. These scores are then converted into weights using exponential scaling: $w_{t+k,i} = \exp(\alpha \cdot r_{t+k}^{(i)})$ where $\alpha$ is a temperature parameter that adjusts the sharpness of the bias. We use these weights to shift the original logits: $\tilde{z}_{t+k,i} = z_{t+k,i} + \beta \cdot w_{t+k,i}$ where $\beta$ is a hyperparameter that modulates the trade-off between specification adherence and the original task objective. Finally, we obtain the action distribution by applying softmax over the adjusted logits: $p_{t+k} = \text{softmax}(\tilde{\mathbf{z}}_{t+k})$

This approach allows for graded preferences that improve flexibility and robustness to dynamics approximation errors. By ensuring that all actions preserve a non-zero probability, the policy remains capable of recovering from errors arising from an imperfect dynamics model. Concretely, if the predicted successor $\hat{x}_{t+1}$ is off by $\epsilon$, an action that appeared marginally unsafe can be safe in the true system, and vice versa. Retaining a weighted down probability for this action gives the sampler a fall-back option whereas HCD would completely rule this action out due to 0 probability. Since we are shifting the probability mass for unsatisfying actions,

it is possible that they are still chosen and lead to a violation. However, this is a tradeoff that we allow to achieve a given task objective. We note that this still ensures higher STL satisfaction than unconstrained actions.

## 4. Evaluation

### 4.1. Implementational Details

We comprehensively evaluate our constrained decoding framework on procedurally generated AI2-THOR (Kolve et al., 2022) indoor scenes with diverse objects and layouts. AI2-THOR is a photorealistic 3D simulation environment built on the Unity engine and is a widely adopted benchmark and training environment for embodied navigation research. We use three state-of-the-art (SOTA) generalist robot navigation policies: Shortest Path Oracle Clone (SPOC) (Ehsani et al., 2024), PoliFormer (Zeng et al., 2025) and Flare (Hu et al., 2025). All three are large transformer-based policies trained on extensive language-conditioned robot trajectory datasets. These models achieve strong zero-shot generalization for a vast variety of navigation tasks that span open vocabulary object-goal navigation ("find a mug"), room-to-room traversal ("visit all rooms"), waypoint-based navigation ("move three meters forward and stop near the red rug"), and attribute-conditioned variants ("locate the chair closest to the refrigerator in the kitchen"). These models also demonstrate reliable zero-shot transfer to real-world environments, achieving robust task satisfaction.

In addition, these models capture three different training paradigms for generalist robot navigation policies. SPOC is trained purely with imitation learning from shortest-path rollouts. PoliFormer employs a hybrid approach that combines reinforcement learning and imitation learning, enabling it to learn long-horizon structure while retaining expert priors. Flare adopts a large-scale pretraining plus fine-tuning on embodied navigation data in line with recent foundation model training paradigms. This diversity in training paradigms allows us to evaluate the applicability of `SafeDec` across different learning regimes.

In this work, we address safety specifications for robotics and, therefore, select those most relevant to real-world deployment. Our specifications are highlighted in Table 1. These specifications are inspired by common safe naviga-

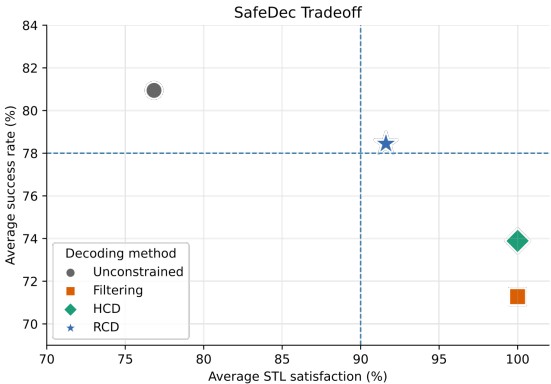

*Figure 2.* **SafeDec trade-off between STL satisfaction and task success.** Average STL satisfaction versus task success rate aggregated across all specifications and three policies. RCD achieves the strongest trade-off, substantially improving STL satisfaction while remaining close to the unconstrained policy's task performance. HCD and filtering achieve near-perfect STL satisfaction, but incur a larger reduction in task success.

tion specifications as highlighted in Menghi et al. (2019) and have varying levels of complexity. These specifications capture state-based invariants ($\phi_{avoid}$, $\phi_{geofence}$), history-dependent conditionals ($\phi_{no\_return}$) and temporal ordering constraints ($\phi_{ordered}$). We encode our test STL specifications using an efficient computational graph-based STL library called STLCG++ that can evaluate multiple state signals in parallel (Kapoor et al., 2025c). This ensures minimal inference overhead at runtime ($10^{-5}$ s per timestep) , which is crucial for policy deployment. For our specifications, we generate random regions in the configuration space at runtime and use their coordinates as spatial predicates. For our dynamics model, we assume a unicycle model, an approximate first-order dynamics abstraction widely used in the robotics literature for analysis and control (Cohen et al., 2024). This representation captures the essential kinematics of motion in the plane and is widely used because it is applicable for diverse robotic platforms.

### 4.2. Experimental setup

We compare our proposed techniques with (1) an unconstrained base model and (2) a base model with a filtering mechanism. The filtering mechanism picks default actions that cannot immediately violate the specification (eg. turning in place) upon predicted violation of the safety specification, similar to the Simplex architecture (Sha, 2001). Simplex architecture is a classic scheme in which a high-performance advanced controller is continuously monitored by a provably safe but less capable backup controller. Simplex based techniques have been used extensively for safety-critical robotics and are a widely accepted standard for runtime-safety comparisons. Additionally, our base policies represent actions as a discrete set of low-level action primi-

tives (e.g., move forward/backward, lateral strafing, rotation, end, and a few other discrete navigation actions exposed by each policy). At each timestep, the policy outputs a categorical distribution over this shared action set. We evaluate performance using two main metrics: STL Satisfaction Rate (**STL St**), defined as the proportion of trajectories that satisfy the specified STL formula, and Task Success Rate (**SR**), which measures standard task success.

### 4.3. Results

Our results are highlighted in Table 2. For brevity, we provide the results for $\phi_{geofence}$ in A.2 since it is syntactically similar to $\phi_{avoid}$. Additionally, we summarize the safety–performance trade-off across all specifications and policies in Figure 2. We also visualize sample trajectories in Figure 3 for one scene and task. Unless stated otherwise, all numbers are averaged over 200 evaluation episodes.

**Do HCD and RCD provide higher STL satisfaction than the unconstrained baselines?** Both HCD and RCD consistently improve STL satisfaction relative to the unconstrained baselines across all models and specifications. All policies with RCD show significant gains, with improvements ranging from 8-24 percentage points. HCD achieves perfect STL satisfaction, with improvements ranging from 16-29 percentage points. We observe that the Simplex-style filtering baseline achieves similar STL satisfaction rate as HCD. This parity is expected, as both methods block any action predicted to violate the specification.

**Which safety enforcement technique achieves task success rates comparable to the unconstrained baselines?** As highlighted in Table 2, Simplex-style filtering attains high STL satisfaction but sacrifices task success because the agent takes predefined safe actions. HCD shows similar behavior: although safety is maximized, success rates are consistently lower than the unconstrained baseline across models and specifications. However, as HCD factors in base model logits, it is able to achieve higher task satisfaction compared to Simplex-style filtering with improvements ranging from 0.5-3 percentage points. In contrast, RCD preserves success rates much closer to the unconstrained level with success-rate drops of 0.5–6.5 percentage points (smaller drop is better). However, we note that RCD does not fully recover success for certain models and specifications, task success remains several points below the unconstrained baseline. Nevertheless, RCD enforces safety while avoiding the large performance penalty observed with filtering.

**Does RCD achieve better task success than HCD while maintaining high STL satisfaction?** While both HCD and RCD improve safety over the unconstrained baseline, they differ in how they balance constraint satisfaction with task

*Table 2.* Comparison by decoding technique across models (SPOC, Flare, PoliFormer) for specifications $\phi_{avoid}$, $\phi_{ordered}$, and $\phi_{no\_return}$. Each cell reports STL satisfaction / success rate (%). Higher is better (↑).

| Decoding | $\phi_{avoid}$: STL St / SR (% ↑) | | | $\phi_{ordered}$: STL St / SR (% ↑) | | | $\phi_{no\_return}$: STL St / SR (% ↑) | | |
|---|---|---|---|---|---|---|---|---|---|
| | SPOC | Flare | PoliFormer | SPOC | Flare | PoliFormer | SPOC | Flare | PoliFormer |
| Unconstrained | 72.0 / 82.5 | 75.5 / 82.0 | 77.0 / 82.5 | 76.0 / 80.0 | 72.0 / 80.5 | 71.0 / 81.5 | 84.0 / 79.5 | 82.0 / 81.0 | 82.0 / 79.0 |
| Filtering | 100.0 / 72.0 | 100.0 / 78.5 | 100.0 / 75.5 | 100.0 / 69.0 | 100.0 / 70.0 | 100.0 / 67.5 | 100.0 / 68.5 | 100.0 / 74.0 | 100.0 / 66.5 |
| HCD | 100.0 / 72.5 | 100.0 / 81.0 | 100.0 / 78.5 | 100.0 / 70.5 | 100.0 / 73.0 | 100.0 / 72.5 | 100.0 / 71.0 | 100.0 / 76.5 | 100.0 / 69.5 |
| RCD | 93.0 / 76.0 | 83.0 / 82.5 | 87.5 / 83.5 | 91.0 / 75.0 | 89.0 / 79.5 | 95.0 / 78.0 | 96.0 / 77.5 | 96.0 / 80.0 | 94.0 / 74.0 |

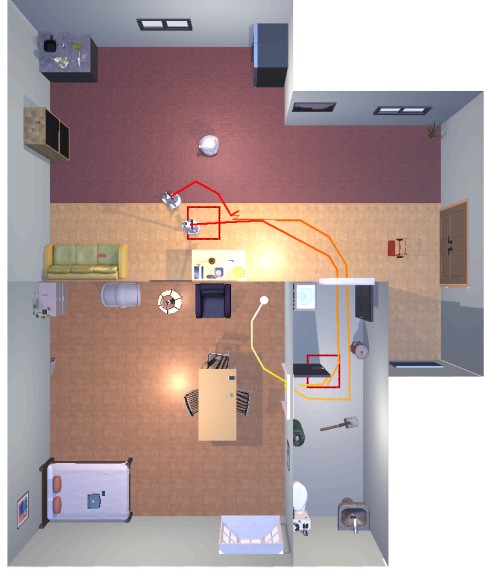
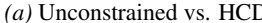

*(a)* Unconstrained vs. HCD

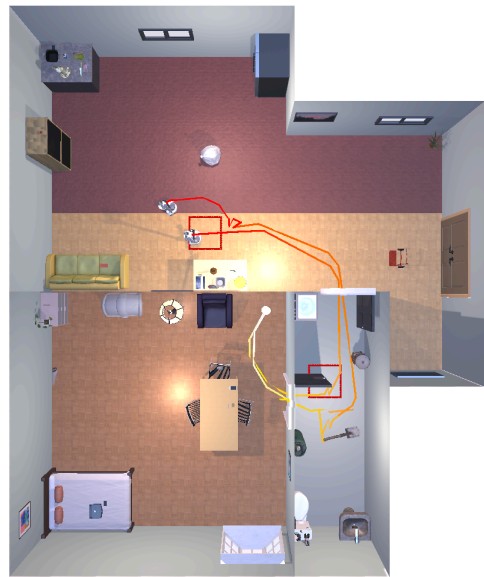

*(b)* Unconstrained vs. RCD

*Figure 3.* Top-down visualization of trajectories for a sample scene under the instruction *"find the sofa"*. The unconstrained model passes through two forbidden regions (red squares) on its way to the sofa (center-left). In contrast, HCD (left) and RCD (right) modify the trajectories to respect STL safety specifications while still reaching the goal. More visualizations are available in Appendix A.1.

success. HCD enforces strict STL satisfaction that results in frequent conservatism and lower successful task completion rates. In contrast, RCD's soft penalization leads to higher task success while still maintaining reasonable STL satisfaction. These results show that RCD achieves a better trade-off between safety and goal-directed behavior, especially in settings where occasional low-risk actions can lead to higher long-term rewards. We additionally performed one-sided two-proportion z-tests with $n = 200$ episodes per cell. Relative to the unconstrained policy, HCD significantly reduces task success in **6/9** conditions ($p < 0.05$), whereas RCD does not significantly reduce task success in any condition. At the same time, RCD significantly improves STL satisfaction over the unconstrained policy in all 9 conditions ($p < 0.05$). These results support the main safety–performance trade-off.

Overall, our proposed techniques effectively enforce safety STL specifications during policy execution. HCD ensures full compliance, but occasionally sacrifices task success due to strict truncation. RCD strikes a balance, offering high

*Table 3.* Comparison with SafeVLA on the SafetyChores benchmark. Lower cost indicates fewer safety violations, while higher success denotes better task completion.

| Method | Cost ↓ | SR (%) ↑ |
|---|---|---|
| ISA (SafeVLA) | 0.205 | 86.5 |
| ISA + HCD | 0.015 | 82.0 |
| ISA + RCD | 0.060 | 86.0 |
| FLARE | 0.192 | 82.0 |
| FLARE + HCD | 0.115 | 84.0 |
| FLARE + RCD | 0.155 | 79.0 |

satisfaction rates and robust performance.

### 4.4. Comparison with safe reinforcement learning baseline

We additionally compare SafeDec with SafeVLA (Zhang et al., 2025), a safe reinforcement learning method that proposes an integrated safety approach (ISA) using explicit safety costs. To enable a fair comparison, we focus on the fragile-object and dangerous-object specifications in

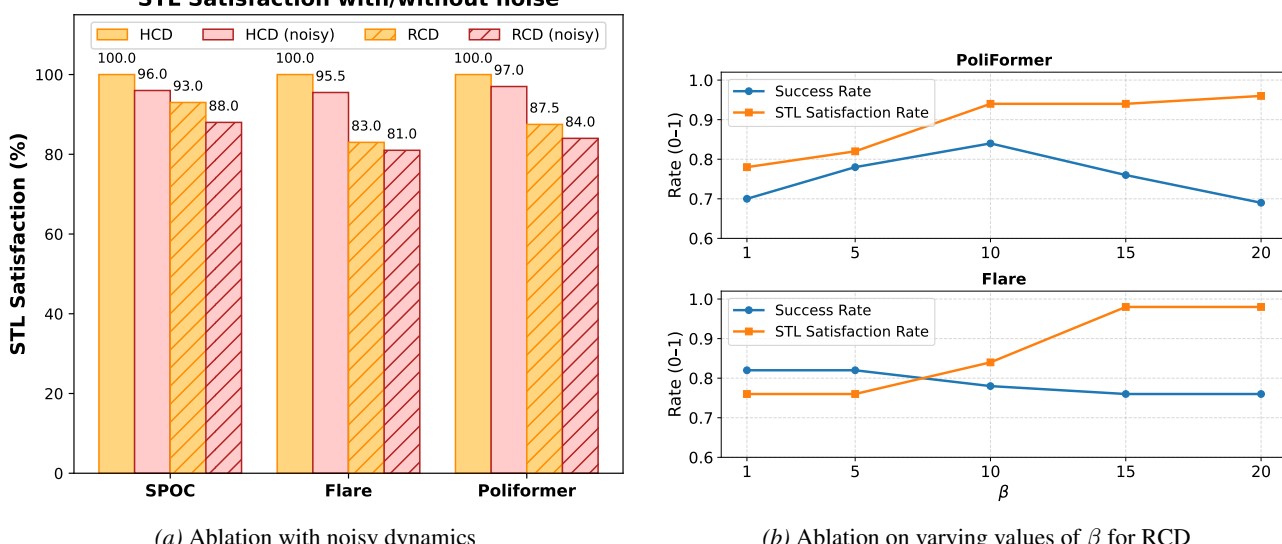

*(a)* Ablation with noisy dynamics

*(b)* Ablation on varying values of $\beta$ for RCD

*Figure 4.* **Ablation studies.** (a) STL satisfaction (%) for HCD and RCD under baseline vs noisy dynamics across base models. (b) Effect of $\beta$ on success rate and safety satisfaction.

the SafetyChores benchmark ([Zhang et al., 2025](#)), which closely align with our setting, and construct corresponding STL specifications for evaluation. We report the composite safety cost defined in SafeVLA, which directly reflects the degree of safety violations across both object classes, along with task success.

Our results are highlighted in Table 3. Our findings indicate that both HCD and RCD substantially reduce safety violations relative to their respective vanilla baselines while largely preserving task success. In particular, integrating HCD with ISA (SafeVLA) achieves an order-of-magnitude reduction in safety cost ($0.205 \rightarrow 0.015$), demonstrating effective enforcement of fragile and dangerous object specifications with minimal impact on task performance. We also find that RCD+ISA preserves task success close to vanilla ISA while reducing safety violations. Our results show that SafeDec, despite being a training-free method, preserves task satisfaction close to the safe reinforcement learning baseline (ISA) for our specifications. Additionally, combining SafeDec with the ISA policy leads to a further reduction in safety costs, highlighting its complementary benefits to data-driven safe learning.

### 4.5. Ablation studies

**Does inaccurate dynamics modeling substantially reduce STL satisfaction?** In this work, we assume a simple unicycle dynamics model due to its generalization capability for diverse robotic platforms. Although this represents a high-level abstraction of true dynamics, such modeling simplifications are standard in the formally assured robot safety literature ([Cohen et al., 2024](#)). However, both RCD and HCD depend on this assumption and inaccurate modeling can impact STL satisfaction. To evaluate the impact of inaccurate dynamics modeling, we conducted an ablation in which we inject Gaussian perturbations into the dynamics (0.01 m per step translational noise i.e. 5% of nominal forward step, 1 ° per step rotational noise i.e. 3.3% of yaw step) for both HCD and RCD. Our results are visualized in Figure 4a. Across all three base models, the drop in STL satisfaction rates from baseline to noisy dynamics is relatively small. We observe that `SafeDec` shows graceful degradation under significant per-step disturbances.

**How do varying values of $\beta$ affect success rate and STL satisfaction in RCD?** As highlighted in section 3.3, RCD uses a hyperparameter $\beta$ that modulates the trade-off between specification adherence and the original task objective. The value of $\beta$ affects the relative weighting between robustness and base logits. To investigate the impact of $\beta$ on the success rate and STL satisfaction, we performed an ablation with varying values for $\beta$ for Flare and PoliFormer. As visualized in Figure 4b, we observe that as $\beta$ increases for PoliFormer, both STL satisfaction and success rate improve in tandem until $\beta = 10$, suggesting that moderate regularization can actually aid policy execution. Beyond this, STL satisfaction continues to improve but at the cost of lower success rates. For Flare, larger $\beta$ values improve STL satisfaction but reduce success rates. These results highlight that the influence of $\beta$ is model-dependent but in general demonstrate that `SafeDec` provides a tunable mechanism to balance safety and performance objectives.

## 5. Related Work

Constraint satisfaction for robotics has been an active area of research that involves techniques such as control barrier functions (CBFs) (Ames et al., 2019), safe reinforcement learning (Gu et al., 2024), and temporal logic-based shielding approaches (Alshiekh et al., 2017). Recently, with the advent of vision language action models and their impressive generalizable capabilities for embodied navigation and other tasks, there are growing concerns about ensuring safety and correctness without retraining these large models. Although classical methods offer formal guarantees, they either require training stage interventions or designing a new classical controller for each safety specification, which can be restrictive. For example, SafeVLA (Zhang et al., 2025) uses safeRL with task-specific safety costs, achieving strong performance in SafetyCHORES tasks. However, the safety specification is expected to be embedded in the training data and loss, meaning that the model cannot generalize to new safety constraints at test time. In contrast, ASIMOV (Sermanet et al., 2025) explores rewriting dangerous instructions with better human-aligned alternatives to steer model behavior without modifying model parameters, but lacks trajectory-level formal guarantees. Our technique achieves a middle ground with the ability to adapt to novel specifications at test time without modifying model parameters while requiring minimal assumptions about the underlying model. There has been some work on logic constrained decoding such as SELP (Wu et al., 2025) that proposes LTL-constrained decoding for language model-based plan generation. However, SELP is unsuitable for STL because its Boolean predicate-based LTL cannot encode numeric bounds (e.g. $||x-x_{goal}|| < 0.1m$) and does not possess quantitative semantics, which is crucial for ranking actions. Moreover, SELP operates at the symbolic planning level, constraining high-level plans produced by LLMs offline, whereas our method enforces safety over per-step low-level actions generated online by embodied policies.

## 6. Limitations and Future Work

In this work, we introduce a constrained decoding framework for enforcing safety specifications for large transformer based robot navigation policies. Our approach enables runtime adaptation to novel safety specifications without retraining. Through experiments across multiple procedurally generated environments, we demonstrated that our method significantly improves STL satisfaction while maintaining high task success rates. Our approach makes two critical assumptions that can be a limiting factor. First, we assume access to specifications that are defined over the state space and that these specifications are generated by roboticists. Although this is a common situation for safety critical deployment contexts like aerial robotics (Aloor et al.,

2023; Luckcuck et al., 2019), these specifications can be difficult to design and involve access to a localization module that can provide accurate state estimation. We plan to overcome this bottleneck by leveraging open-world safety specifications using recent work on embedding spaces-based logic (Kapoor et al., 2025a) and using large language models for generating high level specifications automatically (Li et al., 2025). Second, our approach also assumes access to an approximate dynamics model to evaluate the impact of actions on future trajectories. While a common assumption for provably safe robotics (Cohen et al., 2024), this can limit applicability of our approach. However, it is possible to mitigate this via learned dynamics models such as MoSim (Hao et al., 2025) or world models proposed in Zhou et al. (2025); Micheli et al. (2023) and we plan to explore this in future work.

## Impact Statement

This paper presents work whose goal is to advance the field of Machine Learning. There are many potential societal consequences of our work, none of which we feel must be specifically highlighted here.

## Acknowledgements

We'd like to thank our reviewers for their insightful feedback. This work was supported in part by the National Science Foundation Award CCF- 2144860.

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

# A. Appendix

## A.1. Visualizations

To complement our quantitative results, we provide numerous trajectory plots from evaluations across a diverse set of procedurally generated indoor environments. Figure 5 illustrates representative top-down visualizations of trajectories induced by `SafeDec`. Each plot shows the agent's starting point and the resulting path under constrained decoding, with red circles marking target objects, green boxes denoting forbidden regions, and orange paths depicting the safe trajectories generated by our method. Together with our quantitative analysis, these qualitative results illustrate the performance of `SafeDec` across environments and tasks.

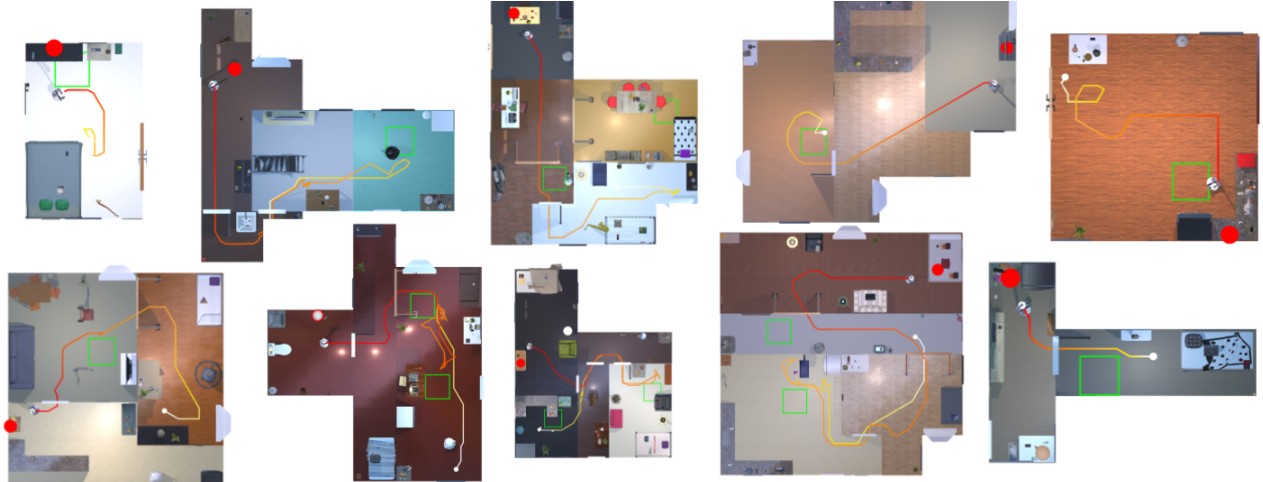

*Figure 5.* Top-down views across indoor environments with `SafeDec` induced trajectories Red circles mark target objects, green boxes are avoid regions, and orange paths show safe trajectories under constrained decoding.

## A.2. Results for $\phi_{geofence}$

*Table 4.* Performance comparison for the geofence specification $\phi_{geofence}$ across decoding techniques and base models. Each cell reports STL satisfaction / success rate (%). Higher is better (↑).

| Decoding | $\phi_{geofence}$: STL St / SR (% ↑) | | |
| --- | --- | --- | --- |
| | **SPOC** | **Flare** | **PoliFormer** |
| Unconstrained | 78.0 / 81.5 | 68.0 / 81.0 | 73.0 / 81.5 |
| Filtering | 100.0 / 72.0 | 100.0 / 66.5 | 100.0 / 67.5 |
| HCD | 100.0 / 76.5 | 100.0 / 67.5 | 100.0 / 72.5 |
| RCD | 95.5 / 80.0 | 80.0 / 71.5 | 85.5 / 77.5 |

