# OpenReview forum: "SafeDec: Constrained Decoding for Safe Autoregressive Generalist Robot Navigation Policies"
_ICML.cc/2026/Conference — ICML 2026 regular_

### Official Review · Reviewer_jHas · 2026-02-26

**Soundness:** 3
**Presentation:** 4
**Significance:** 2
**Originality:** 2
**Overall Recommendation:** 4
**Confidence:** 4

**Summary:**

The paper tackles the question of safe decoding of actions from transformer models. Specifically, it operates in a setting where an embodied agent, instantiated as a transformer model, has to generate actions over a fixed future time horizon such that after they are executed in the environment, the obtained state trajectory conforms to certain behavior and safety constraints, described as Signal Temporal Logic (STL) formulas.

To guarantee that the actions conform to such safety constraints, the authors use techniques based on constrained decoding. First, the predicted action sequence is rolled-out using a basic, surrogate dynamics model, obtaining the state trajectory resulting from those actions. Subsequently, given that actions are discrete, a robustness score is computed for each candidate action in each timestep of the future horizon. This score is then normalized and added to that action's logits. Everything else that follows (the softmax that computes a probability over actions) is as usual (without modifications).

Evaluation is done in the AI2-THOR environments, over 3 models - SPOC, PoliFormer, and Flare. Results suggest that the proposed constrained decoding technique, called Robustness Constrained Decoding (RCD), satisfies the majority of constraints while keeping the success rate relatively high.

**Compliance With Llm Reviewing Policy:**

Affirmed.

**Key Questions For Authors:**

1. You say that STL can express temporal ordering or conditional safety relationships. What are some examples of behavior that cannot be expressed in STL? In general, are STLs popular in practice? Are they used extensively in the real-world?
2. In line 88 you mention "liveness operators". What are these?
3. Line 102, grammatical error "continuous time real-valued".
4. You say that manipulation after sampling "undermines the inductive biases learned during training". How can this statement be proved experimentally?
5.  "due it's" should be "due to its"
6. Regarding experiments, could you plot the trade-off between the task success rate and the safety satisfaction rate? It'd be nice to see visually how much we're sacrificing one for the other.

Overall, it's a good paper. Even though the contribution itself is rather small, I'm rating it "weak accept" because the paper is accessible, understandable, and this direction of safe action decoding will be useful in the future.

**Limitations:**

yes

**Strengths And Weaknesses:**

Strengths
===
1. Overall, an exceptionally well-written paper and a joy to read. Technical descriptions were clear, background information was useful, and explanations were well-organized and positioned, so that they come just when the reader needs them.
2. The method itself, Robustness Constrained Decoding (RCD), is very simple and intuitive. It does what it sets out to do - if a continuous score for the achievement of the desired behavior is available, RCD is a way to select actions so as to maximize this score.

Weaknesses
===
1. The method itself, RFC, while simple and easy to implement, seems rather incremental. The reason is that it offloads the important thing - the computing of the robustness score - to the STL library. The robustness score could be substituted with a reward estimated from a learned critic, or from some hardcoded constraint, or any other number, and nothing will change technically. Once the score is available, linearly shifting the logits based on this score is a trivial modification.
2. The two limitations mentioned - that the STL safety constraints need to be provided beforehand, and that the dynamics model is too basic (could be more realistic) - are bigger challenges worth investigating. It would have been nice if any of these limitations was solved, so as to make a stronger contribution. Otherwise, it seems that you're relying on the provided dynamics to simulate the resulting states, and the STL engine to evaluate the behavior, and then you're only shifting the logits. Perhaps adding more motivation,  in terms of what this change enables, will be useful to help make contribution stand on its own.

---

> ### Author Rebuttal · Authors · 2026-03-30
>
> # Response to Reviewer jHas
>
> We thank Reviewer 4 for the thorough and encouraging review.
>
> **A note on novelty**: SafeDec's novelty is not in its components but in what they enable. First, new capability: it enforces user-specified temporal logic constraints on pretrained transformer policies at inference time with no retraining, no offline compilation, and no model modification something no prior method provides. Second, new problem formulation: framing robot policy safety as a decoding problem is a non-trivial step, and one that makes the composition meaningful.
>
> Now, We address the weakness and each question below.
>
> ## **W1 — On substitutability of the score**
>
>
>  The reviewer is correct that, mechanically, any scalar signal could shift logits. The contribution of RCD is not the shift itself, but the argument for *why STL robustness is the right signal* in this setting. Unlike a learned critic, STL robustness: (i) requires no training and adapts to any user-specified constraint at deployment time; (ii) has formally grounded semantics — the score's magnitude carries interpretable meaning tied to the specification; (iii) captures *temporal structure* across trajectory history, not just pointwise state quality. A reward from a learned critic must be retrained per specification; a hardcoded constraint lacks temporal compositionality. The insight that STL robustness can serve as a logit-shaping signal for pretrained transformer policies, sufficient for meaningful safety gains with no model modification, no training, and no offline compilation, is itself a contribution.
>
> ## **W2 — Dynamics model assumption and predefined specifications**:
>
> We agree that learned specifications and richer dynamics are the more challenging open problems, and frame them explicitly as future work in Section 6. SafeDec is intentionally designed as a lightweight, deployable baseline that cleanly isolates the inference-time constrained decoding problem. Our simple dynamics models also allow transferability to various navigation robot morphologies and is a common practice for real world safety deployment contexts such as in ViSafe (Kapoor et al, 2025). Establishing this baseline and demonstrating it works across three policies, four specifications, and 200 environments with no retraining is a necessary step before tackling the harder problems the reviewer identifies. We will strengthen the introduction to make this positioning explicit.
>
> ## **Q1 — What Cannot Be Expressed in STL? Are STL Specifications Used in Practice?**
>
> **Expressivity limits.** Properties outside STL's expressive power include:
>
> - **Probabilistic/stochastic properties** — e.g., "the robot reaches the goal with probability > 0.9" (requires PCTL).
> - **Unbounded liveness** — e.g., "the robot *eventually* reaches the goal" without a time bound. STL requires finite time intervals.
> - **Fairness and infinite-horizon properties** — e.g., "the robot visits region A and B infinitely often" (requires LTL over infinite traces).
> - **Epistemic or belief-level properties** — properties over the robot's internal belief state rather than observable signals.
>
> **Practical use.** STL is widely used in safety-critical robotics: autonomous driving lane-keeping and safety distances (Veer et al, 2022) and aerial robotics (Aloor et al., 2023). The formal methods in learning survey (Manganaris et al, 2025) and formal verification survey (Luckcuck et al., 2019) document extensive real-world adoption.
>
> ## **Q2 — Liveness Operators**
>
> Liveness operators encode goal-reaching: *eventually* (F, "at some point in [a,b] the condition holds") and *until* (U, "condition A holds until B becomes true"). These specify that *something good eventually happens*, as opposed to safety operators (G) which are used to specify that *nothing bad ever happens* (Alpern & Schneider, 1987).
>
> ## **Q3 — Grammatical Error: "continuous time real-valued" (Line 102)**
>
> Thank you. This will be corrected to "continuous-time, real-valued" in the final version.
>
> ## **Q4 — "Undermines Inductive Biases" — Can This Be Proved Experimentally?**
>
> The claim is supported by comparing HCD against the Simplex-style filtering baseline in Table 2. Both methods block actions predicted to violate the specification, but filtering defaults to a fixed safe action (rotation in place), completely overriding the model's distribution, whereas HCD retains the model's logit ordering over the remaining safe actions. HCD consistently achieves 0.5–3 pp higher task success than filtering despite identical STL satisfaction — attributable to HCD preserving the policy's inductive biases within the safe action set. We will make this argument explicit in the Table 2 discussion.
>
> ## **Q5 — Typo: "due it's" (Section 4.5)**
>
> Thank you. It will be corrected to "due to its" in the final version.
>
> ## **Q6 — Trade-off Plot (Safety vs. Task Success)**
>
> Thank you for this suggestion. We will include this figure in the appendix of the final version.

---

> > ### Author Rebuttal · Reviewer_jHas · 2026-04-03
> >
> > Thank you for your clarifications.
> >
> > I think my questions are fully-resolved in the sense that the rebuttal's clarifications do provide more motivation and reasoning behind the design of the proposed STL approach.  However, some answers, such as that to Q1, are excessively theoretical in nature and do not manage to convince me or increase my confidence in the approach's utility.
> >
> > Overall, I will keep my score as that in the initial review - "Weak accept".

---

> > > ### Author Response · Authors · 2026-04-08
> > >
> > > We thank the reviewer for this suggestion and their deep engagement. We agree that our earlier response to Q1 leaned too much on formal expressivity and not enough on **concrete robotics use cases**. We provide a more grounded answer for STL usage.
> > >
> > > **Why STL is useful in practice.** In robotics, STL is valuable because it lets one write the kinds of rules that operators, safety engineers, and regulators actually care about. Examples include:
> > >
> > > - **Autonomous driving:**
> > >   “always keep a safe following distance,”
> > >   “always remain within the drivable region / lane boundaries,”
> > >   “do not exceed the speed limit,”
> > >   “if a lead vehicle brakes, maintain safe separation for the next few seconds.”
> > >   These are real world examples where STL and closely related temporal logics have been used to formalize traffic rules, safety distance rules, and automated-driving decision/monitoring requirements. [1-2]
> > >
> > > - **Aerial robotics / airfield operations:**
> > >   “avoid no-fly regions at all times,”
> > >   “remain above a minimum altitude except during landing,”
> > >   “approach the runway only after entering the correct traffic pattern,”
> > >   “after crossing a waypoint, do not re-enter the restricted zone.”
> > >  Prior work used STL online for general-aviation aircraft planning around a non-towered airfield, where rules were encoded as spatio-temporal constraints and used during decision making. [3]
> > >
> > > - **Manipulation / mobile manipulation:**
> > >   “reach object A before object B,”
> > >   “keep the gripper outside unsafe regions while approaching the target,”
> > >   “eventually reach the target pose while always avoiding collision regions,”
> > >   “complete a pick-and-place sequence within a time bound.”
> > >   STL robustness has been used in robotic manipulation as a task/specification signal precisely because these tasks involve temporal ordering and safety conditions over trajectories, not just single-state rewards. [4-5]
> > >
> > > These examples are the reason we use STL rather than a generic scalar score. A learned reward or critic can assign a number to a state/action, but STL provides a way to specify **human-readable rules over trajectories**.
> > >
> > > We hope this helps contextualize STL’s usage in real world robotics. If our answers help position the work better, please consider adjusting the score!
> > >
> > >  - [1] Mohammad Hekmatnejad et al. Encoding and monitoring responsibility sensitive safety rules for automated vehicles in signal temporal logic, MEMOCODE 2019
> > > - [2] Meng, Yue et al.. "Diverse controllable diffusion policy with signal temporal logic." IEEE Robotics and Automation Letters, 2025
> > > - [3] Aloor, Jasmine Jerry, et al. "Follow the rules: Online signal temporal logic tree search for guided imitation learning in stochastic domains." ICRA, 2023.
> > > - [4] P. Aniruddh Gopinath, et al., “Learning from demonstrations using signal temporal logic,” in CoRL, 2020.
> > > - [5] A. G. Puranic, et. al, “Learning From Demonstrations Using Signal Temporal Logic in Stochastic and Continuous Domains,” IEEE Robotics and Automation Letters, 2021

---

### Official Review · Reviewer_h1SK · 2026-03-12

**Soundness:** 3
**Presentation:** 3
**Significance:** 3
**Originality:** 3
**Overall Recommendation:** 4
**Confidence:** 4

**Summary:**

This paper proposes SafeDec, an inference-time constrained decoding framework for autoregressive robot navigation policies under STL safety specifications. The method includes HCD, which hard-masks unsafe actions, and RCD, which uses STL robustness to softly bias decoding toward safer actions while preserving task performance. Experiments on three navigation policies show that HCD achieves perfect STL satisfaction on the reported benchmarks, while RCD often provides a better safety-success tradeoff than hard masking or simple filtering

**Compliance With Llm Reviewing Policy:**

Affirmed.

**Final Justification:**

The paper presents a technically solid and interesting approach to improving the safety of pretrained robot navigation policies through STL-guided constrained decoding. The core idea is original in this setting, and the experiments across multiple base policies show meaningful safety gains with a useful safety-performance tradeoff. Overall, I find the work sound and relevant to the safe embodied AI community.

The rebuttal addressed my main technical concerns well. In particular, it clarified the online prefix-based STL evaluation, the distinction between the guarantees of HCD and RCD, and the practical choice of the beta parameter. These responses increased my confidence in the paper’s clarity and soundness. However, my main reservation remains that the evaluation is limited to simulation, which constrains the paper’s demonstrated significance. As a result, the rebuttal reinforced rather than changed my overall assessment, and I keep my final recommendation at 4 (Weak Accept).

**Key Questions For Authors:**

- Can the authors clearly separate the guarantees of HCD and RCD? In particular, is formal STL satisfaction claimed only for HCD, or for the full SafeDec framework?
- How should the beta parameter in RCD be selected in practice? If it is task or model-dependent, can the authors provide tuning guidance or transferability evidence?
- Have the authors considered a mixed setting where some constraints are enforced hard and others are softened, since in many real applications certain constraints should never be violated?

**Limitations:**

yes

**Strengths And Weaknesses:**

Strengths

- The paper addresses an important problem: adding explicit safety control to pretrained embodied policies without retraining.
- The core idea is interesting and reasonably novel in this setting: applying STL-guided constrained decoding to low-level robot action generation, with both hard (HCD) and soft (RCD) variants.
- The empirical evaluation is fairly convincing for a first study: the method is tested on three strong base policies and consistently improves safety over unconstrained decoding.
- RCD is a useful contribution in its own right, since Table 2 shows a meaningful tradeoff between task completion and constraint satisfaction, and the comparison with SafeVLA suggests the approach is complementary to training-time safe RL.


Weaknesses


- The formal presentation is incomplete in places, especially regarding how STL violations are checked online for partial decoded prefixes and how operators such as U are handled.
- The experimental setting is somewhat limited: results are simulation-only and mostly use randomly generated spatial constraints, so the real-world significance for safety-critical deployment is still uncertain.
- Some practical details remain unclear, including the mismatch between the broad multimodal problem statement and the evaluated setup, and how the beta parameter in RCD should be chosen in practice.

---

> ### Author Rebuttal · Authors · 2026-03-30
>
> # Response to Reviewer h1SK
>
> We thank Reviewer 3 for the positive and constructive assessment. We address all questions and noted weaknesses below.
>
> ---
>
> ## **W1 — Incomplete Formal Presentation (Partial Prefixes, Until Operator)**
>
> We appreciate this observation. We clarify:
>
> **Partial decoded prefixes:** At each decision step t, SafeDec evaluates the STL specification over the trajectory prefix x_0:t-1 already executed plus the  predicted next states for each candidate action. For safety specifications (which are invariants or conditional avoidances), this prefix-based evaluation is exact: a trajectory violates a safety property as soon as the prefix violates it. The history is maintained as an online running buffer and passed to STLCG++ at each step.
>
> **Until operator (U):** For ϕ_ordered (which uses a weak-until formulation G(¬R2) ∨ (¬R2 U R1)), the robustness at each step is evaluated over the accumulated prefix. Since we focus on safety specifications (Alpern & Schneider, 1987), the liveness component (eventually reaching R1) is not enforced by the constraint: the formula encodes "avoid R2 forever, or avoid R2 until R1 is reached," which is a purely safety-type condition. We will add a clarifying paragraph to Section 3.2 on prefix evaluation and the scope of the Until operator.
>
> ---
>
> ## **W2 — Limited Experimental Setting (Simulation-Only, Random Spatial Constraints)**
>
> We acknowledge that the evaluation is simulation-only. This is standard practice for initial studies on constrained decoding for robot policies given the high cost of real-world evaluation at scale (200 episodes per condition). The procedurally generated environments span diverse room layouts, object configurations, and navigation contexts, providing strong coverage. We note that the three base policies (SPOC, PoliFormer, Flare) have themselves demonstrated reliable zero-shot transfer to real-world environments (Zeng et al., 2025), and the safety enforcement mechanism operates independently of the simulation, suggesting good real-world transferability. We agree that real-world validation is an important future step and will note this explicitly.
>
> ---
>
> ## **W3 — Beta Parameter Selection**
>
> Based on our ablation (Figure 3b), β = 10 is a reasonable default that achieves a good tradeoff across both tested policies. In general, we recommend:
>
> - Start with β = 10.
> - If STL satisfaction is insufficient, increase β (at the cost of task success).
> - If task success is unacceptably low, decrease β.
>
> The influence is model-dependent: PoliFormer shows a joint improvement in both safety and success up to β = 10 (suggesting the base policy benefits from STL guidance), while Flare shows a monotone tradeoff. In practice, a small validation sweep over {1, 5, 10, 15, 20} on 20–30 episodes is sufficient to identify a suitable value. We will add this guidance to the appendix.
>
> ---
>
> ## **Q1 — Formal Guarantees of HCD vs. RCD**
>
> To be precise:
>
> - **HCD** provides a formal STL satisfaction guarantee *under the assumed dynamics model*: if the unicycle model is exact, no executed trajectory will violate the specification. This guarantee is conditional on the dynamics assumption.
> - **RCD** does not provide a hard guarantee. It soft-guides the action distribution toward higher-robustness actions while preserving non-zero probability for all actions. It is possible that a low-robustness action is still sampled and leads to a violation, which is the tradeoff that allows RCD to maintain higher task success.
>
> We will add a dedicated remark in Section 3 to make this distinction explicit and will ensure the abstract and contributions section clearly attributes formal guarantees only to HCD.
>
> ---
>
> ## **Q2 — Beta Selection (see Weakness response above)**
>
> ---
>
> ## **Q3 — Mixed Hard/Soft Constraints**
>
> Thank you for this suggestion. The current architecture already supports this naturally: HCD masking and RCD soft-weighting operate independently on per-action logits, so it is straightforward to apply HCD to a subset of safety-critical predicates (e.g., ϕ_avoid for physical obstacles) and RCD to lower-priority constraints (e.g., ϕ_ordered for ordering preferences). We will include this as an explicit extension in Section 6 and note it as a priority for future work.

---

> > ### Author Rebuttal · Reviewer_h1SK · 2026-04-04
> >
> > I acknowledge that I have read the author rebuttal. The rebuttal adequately addresses my main concerns, so I would keep my score at 4, because the evaluation remains limited to simulation. I would consider a higher score if the paper included real-world experiments or additional metrics in a more realistic simulator.

---

> > > ### Author Response · Authors · 2026-04-08
> > >
> > > We thank the reviewer for their feedback and appreciation. We agree that simulation-only evaluation is a limitation, and we do not claim that it fully substitutes for real-world validation. Our point is narrower: **the AI2THOR simulation study is still informative for this setting because SafeDec is evaluated on pretrained navigation policies that have themselves already demonstrated strong sim-to-real transfer**. In particular, **SPOC was also trained on AI2-THOR data** and has shown to transfer to the real world with no visual adaptation or real-world finetuning; the paper reports 88 real-world trials across two physical environments and notes that the simulation-to-real performance drop is small overall and minimal for navigation tasks. Likewise, **Poliformer is explicitly presented as a transformer policy trained in AI2-THOR at scale that can navigate in the real world without adaptation**, with reported sim-to-real zero-shot gains over prior baselines on both LoCoBot and Stretch RE-1 embodiments.
> > >
> > > This matters for our paper because SafeDec does **not** retrain a new policy in simulation; rather, it is a lightweight inference-time decoding mechanism layered on top of base policies whose real-world transfer has already been established. The safety intervention itself is also not tied to simulator-specific supervision signals, but to online evaluation of STL rules over the robot state and an approximate dynamics model. Thus, while a real-robot evaluation would certainly strengthen the paper, we believe our current experimental setting is still meaningful: it evaluates whether temporal-logic-guided decoding can improve safety for exactly the class of pretrained embodied policies that prior work has already shown to transfer beyond simulation.
> > >
> > > If our answers help position the work better, please consider adjusting the score!

---

### Official Review · Reviewer_stxi · 2026-03-13

**Soundness:** 2
**Presentation:** 3
**Significance:** 2
**Originality:** 1
**Overall Recommendation:** 2
**Confidence:** 4

**Summary:**

The paper proposes SafeDec, an inference-time constrained decoding framework that enforces formal safety specifications in transformer-based robot policies without requiring training. Safety requirements are expressed in Signal Temporal Logic (STL) and enforced by intervening at the action decoding stage using an approximate dynamics model. Specifically, the authors propose two ways of modifying the action selection process: (i) Hard Constrained Decoding (HCD), which masks unsafe actions, and (ii) Robust Constrained Decoding (RCD), which reweights action logits according to STL robustness scores. Experiments are conducted in AI2-THOR navigation environments with multiple safety specifications. The presented results show that the incorporation of the STL safety filter can improve safety/specification satisfaction without significantly degrading task success rates. Overall, SafeDec demonstrates that formal safety constraints can be enforced for large transformer policies purely through inference-time decoding control.

**Compliance With Llm Reviewing Policy:**

Affirmed.

**Final Justification:**

Thank you very much for the response. My primary concern about the novelty still remains. And the major limitation of requiring the dynamics model, state information and discrete action poses significant limitations to generalization. Based on this I selected my final score.

**Key Questions For Authors:**

**Questions:**
1. Does the method with one-step lookahead work for stationary agents? Additional experiments could help clarify this.
2. Why does the success rate drop in most cases with the integration of the STL checker compared with the unconstrained case instead of increasing?
3. Why is this approach justifiable over learning policies for general STL specifications as in [1], which are model-free and not limited to safety?

[1] Yue Meng, Fei Chen, and Chuchu Fan. TGPO: Temporal Grounded Policy Optimization for Signal Temporal Logic Tasks. arXiv preprint arXiv:2510.00225, 2025.

**Limitations:**

Yes

**Strengths And Weaknesses:**

**Strengths:**
1. The paper is well written, and the framework is presented clearly with results, making it easy to follow.
2. SafeDec enforces safety constraints at test time without requiring retraining or modification of the base policy.
3. The method operates independently of the underlying policy architecture and is demonstrated across multiple policies (SPOC, Flare, PoliFormer).
4. Experiments show the approach works across several state-of-the-art robot navigation models. The presented results show that the STL filter achieves significant gains in specification satisfaction, but at the expense of success rates in most cases across all tested models.

**Weaknesses:**
1. The main weakness is the novelty claim of the paper. SafeDec adapts constrained decoding techniques from NLP to enforce formal safety constraints on transformer-based robot navigation policies at inference time.
2. The safety monitor requires access to the true system and environment states, which poses a major practical limitation.
3. The requirement of an approximate dynamics model outweighs the benefit of easy integration with arbitrary transformer-based policies.
4. The proposed framework only works in a discrete action space setting.
5. STL predicates must be defined in state space (e.g., spatial regions), requiring prior environment knowledge or maps.

---

> ### Author Rebuttal · Authors · 2026-03-30
>
> # Response to Reviewer stxi
>
> We thank Reviewer 2 for their insightful feedback. We address all weaknesses and questions below.
>
> ---
>
> ## **W1 — Novelty**
>
> SafeDec's contribution is not merely adapting constrained decoding from NLP. The key challenge (Section 3.1) is that **constraint checking for robot policies cannot be performed in token space** : unlike NLP constraints (regex, grammar), safety specifications over dynamical systems are defined over state variables and require a dynamics stepping function to evaluate. This requires a new problem formulation, integration of STL robustness into logit reweighting (RCD), and a dynamics-grounded lookahead. No prior work combines training-free decoding, STL quantitative semantics, and online constraint checking for transformer-based robot policies. Please also see our note to novelty in response to reviewer jHas.
>
> ---
>
> ## **W2 — Access to True System and Environment States**
>
> SafeDec does **not** require access to hidden environment state — only the robot's own pose (x, z coordinates), available from standard localization. This is a common assumption in safety-critical robotics and no more restrictive than classical safe controllers operating on the same platforms.
>
> ---
>
> ## **W3 — Dynamics Model Requirement**
>
> The dynamics model in SafeDec is a **lightweight first-order abstraction** (e.g., a unicycle model), widely applicable across mobile robot morphologies, requiring no task-specific training or system identification. Its role is limited to short-horizon prediction for constraint evaluation, not precise long-term simulation. As shown in our ablation (Figure 3a), STL satisfaction degrades only marginally under significant per-step noise, confirming robustness to modeling inaccuracies. The framework is also **agnostic to the dynamics model choice** — learned world models can be used as drop-in replacements.
>
> ---
>
> ## **W4 — Discrete Action Spaces Only**
>
> We acknowledge this limitation. However, discrete action spaces are extremely common in modern embodied AI and VLA policies, including PACT [1], RT-2 [2], and policies using the FAST tokenizer [3], all of which discretize continuous control into token-level actions for transformer-based sequence modeling.
>
> [1] Bonatti et al., IROS 2023. [2] Zitkovich et al., CoRL 2023. [3] Pertsch et al., arXiv 2501.09747.
>
> ---
>
> ## **W5 — Prior Environment Knowledge for STL Predicates**
>
> We acknowledge this limitation and have noted it in Section 6. This assumption is **common in safety-critical deployment settings**, where constraints (geofences, no-go zones, proximity limits) are derived from maps, perception, or regulatory requirements. SafeDec is compatible with two extensions to relax this: (1) learned safety specifications [4], and (2) automatic specification generation from natural language via LLMs [5].
>
> [4] Nakamura et al., RSS 2025. [5] Chen et al., arXiv 2305.07766.
>
> ---
>
> ## **Q1 — Does One-Step Lookahead Work for Stationary Agents?**
>
> SafeDec makes no assumption about whether the agent is stationary. It evaluates candidate actions by their predicted next states under the dynamics model, regardless of motion magnitude. As long as an action induces a state transition (which holds for all discrete actions used), the lookahead is well-defined. If we misinterpreted the question, please clarify and we will run those experiments as requested.
>
> ---
>
> ## **Q2 —Why Does Success Rate Drop with the STL Checker?**
>
> This is addressed in Section 4.3. HCD hard-masks actions predicted to violate the specification; when the policy's preferred forward-progress actions lead toward forbidden regions, only lower-utility actions (e.g., rotations) remain and get chosen. Hence, the policy sometimes doesnt reach the goal within episode horizon (hence lower success rates). Additionally, some environments are inherently infeasible: randomly generated forbidden regions can block all paths to the goal, causing failure regardless of the policy. RCD substantially mitigates this: success rate losses under RCD are 0.5–6.5 pp vs. 1–13 pp for HCD. The safety–performance tradeoff is characterized in the β-ablation (Figure 3b).
>
> ---
>
> ## **Q3 — Why Is This Approach Justifiable over Learning Policies for General STL Specifications (TGPO)?**
>
> Please also see our response to W1 Reviewer evwF. TGPO requires **full RL retraining** with STL-shaped rewards. SafeDec targets a complementary scenario where: (1) **retraining is prohibitively expensive**: base policies are large generalist models with zero-shot real-world transfer; (2) **the safety specification is unknown at training time**:  SafeDec adapts to novel specs at deployment without any model update; (3) **generalization must be preserved**: retraining on specific constraints risks degrading zero-shot capabilities. Table 3 further shows SafeDec combines with safe RL baselines (ISA from SafeVLA) for additional gain: the methods are **complementary**, not competing.

---

> > ### Author Rebuttal · Reviewer_stxi · 2026-04-04
> >
> > Thank you very much for the response. My primary concern about the novelty still remains. And the major limitation of requiring the dynamics model, state information and discrete action poses significant limitations to generalization. Based on this I selected my final score.

---

> > > ### Author Response · Authors · 2026-04-08
> > >
> > > We appreciate the reviewer's response to our rebuttal. As mentioned earlier SafeDec's novelty is not in its components but in what they enable. First, new capability: it enforces user-specified temporal logic constraints on pretrained transformer policies at inference time with no retraining, no offline compilation, and no model modification something no prior method provides. Second, new problem formulation: framing robot policy safety as a decoding problem is a non-trivial step, and one that makes the composition meaningful.
> > >
> > > Second, Our claim is **not** that SafeDec is a universally applicable safety framework for all robot policies. Rather, it targets an important and increasingly common regime: **pretrained autoregressive embodied policies with discrete tokenized actions**, where retraining is expensive and safety rules may need to change at deployment time. In this regime, we believe the contribution is meaningful because SafeDec provides a way to enforce new temporal-logic rules.
> > >
> > > We also agree that the method relies on assumptions. However, these assumptions are weaker than learning a new safe policy from scratch. More importantly, these assumptions should be viewed relative to the problem we address. Existing alternatives such as STL-constrained RL or reward-shaped policy learning typically require retraining for each specification or task family, whereas SafeDec is designed for the complementary case where the **base policy is already trained** and the safety rule is introduced or revised **after training**. That deployment-time adaptability is the main reason we believe this setting is worth studying. This sentiment is echoed by all reviewers.
> > >
> > > We will revise the paper to make this narrower claim explicit: SafeDec is not intended to replace general safe control or safe RL, but to provide a practical inference-time safety layer for pretrained embodied transformer policies under lightweight and standard robotics assumptions.
> > >
> > > Given the strengths of the technique highlighted by the reviewer, I hope they consider raising the score!

---

### Official Review · Reviewer_evwF · 2026-03-20

**Soundness:** 2
**Presentation:** 3
**Significance:** 2
**Originality:** 2
**Overall Recommendation:** 4
**Confidence:** 4

**Summary:**

This paper deals with safe decoding problems of robotics model, e.g., SPOC, Flare.
Actions violating STL constraints are either masked (HCD) or soft-penalized, without retraining the base policy.
Constraint checking relies on a model predictive control-like approach, one-step lookahead, under the assumption that the model is given.
STL-violating tokens are masked, or their probabilities are distorted low enough, so that the action tokens are not sampled.

**Compliance With Llm Reviewing Policy:**

Affirmed.

**Final Justification:**

I think reducing the cost of safety from the simple approach is quite beneficial in robotics and the embodied control problem, though there are still concerns about the rule-making costs. As neuro-symbolic approaches are getting attentio especially for the embodied control problem, the low-cost real-time decision making will make beneficial for this area.

Though I have no objection if the final decision maker decides to reject the paper

The concerns are still valid with respect to the rule-making costs, and somehow, one may still view it as an incremental approach. In an abstract sense, it is one of the shielding approaches which are prominent in Safe RL literature (e.g., [a-c]).

In this sense, I considered a novelty from temporal logic's benefits of low costs, besides of the rule-generating costs.

**Key Questions For Authors:**

**Q.1.** (Besides the violation of STL), what is the rule of failure and what causes failure for the SafeDec policy? Did incorrect paths accumulate, or did something like a collision occur?

**Q.2.** In the Appendix visualization (top row, fourth figure from the left), the robot seems to start inside or on the boundary of the violation area (green box). Also, in the ablation study, even with very small noise (5% translational, 3.3% rotational), HCD shows about a 4% violation. Can you specify the generation rules for the starting point and the violation area? I am curious about the rule for excluding generation inside the violation area.

**Limitations:**

Yes

**Strengths And Weaknesses:**

### **Strength**
The approach is simle: it is training free, model-agnostic and low overhead.
The paper is easy to follow and well-structured.

### **Weakness**

**Weakness 1** Novelty

I have concerns about the novelty of this paper. As the authors discussed in the related work, there has been research about the LTL constraint approach like SELP [1]. Though the authors have argued that SELP can only deal with boolean logic because of its LTL basis, the difference in the fundamental mechanism has not been discussed in the related work.

Further, there are several papers addressing similar problems [2, 3].
For example, in [2], the same problem has been addressed with both model-predictive control and a learned approach. While in [3], the problem is addressed with an (inference-time) gradient approach, modifying the action by
$a \gets a + \nabla_{a} \text{soft violation}(a)  $. In [4], the RL learning problem combined with STL logic has been addressed.

The authors should argue their novelty and contributions within this literature in the related work section


**Weakness 2** Single environment experiments

The authors conducted experiments only on the navigation task in a single AI2-Thor environment. As shown in [1] with driving with safe rules, or in [5] with manipulation considering object states, safety tasks are quite diverse, and it is difficult to consider the proposed method verified solely through a simple obstacle-avoiding navigation task. I am not arguing that the authors should conduct all the experiments shown in [1] and [5]. However, the fact that there is only a single experimental task is a valid concern. Please provide additional experiments that demonstrate the diversity of safety.

**Weakness 3 (Minor)** Significant of RCD (vs HCD)

The authors proposed RCD to increase the success rate while allowing for slight STL violations. I ran a one-sided proportion z-test and a chi-square test on Table 2, and with a few exceptions (SPOC  $\phi_{no\\_return}$, $p=0.069$, one-side proportion) I obtained sufficiently large p-values ($p > 0.1$).

Given that RCD's success rate is higher than HCD's across all results, it is convincing that RCD improves the success rate. However, this does not seem to be a sufficient reason to adopt RCD, which increases the success rate with statistical uncertainty, while abandoning HCD, which lowers the STL violation rate with near statistical certainty.

Please discuss about p-value about success rate and constraint violation; e.g., the authors may discuss the success rate between RCD or HCD so that the statistical difference is not so large, therefore, it is an improvement in violation rate without harming the success rate significantly.




### **Refs.**
[1] Wu, Yi, et al. "SELP: Generating safe and efficient task plans for robot agents with large language models." 2025 IEEE International Conference on Robotics and Automation (ICRA). IEEE, 2025.

[2] Meng, Yue, and Chuchu Fan. "Signal temporal logic neural predictive control." IEEE Robotics and Automation Letters 8.11 (2023): 7719-7726.

[3] Sun, Yang, et al. "Redriver: Runtime enforcement for autonomous vehicles." Proceedings of the IEEE/ACM 46th International Conference on Software Engineering. 2024.

[4] Meng, Yue, Fei Chen, and Chuchu Fan. "TGPO: Temporal Grounded Policy Optimization for Signal Temporal Logic Tasks." arXiv preprint arXiv:2510.00225 (2025).

[5] Ahn, Sanghyun, et al. "Towards reliable code-as-policies: A neuro-symbolic framework for embodied task planning." arXiv preprint arXiv:2510.21302 (2025).

---

> ### Author Rebuttal · Authors · 2026-03-30
>
> # Response to Reviewer evwF
>
> We thank the reviewer for the detailed feedback. We address each concern below.
>
> ---
>
> ## **W1 — Novelty**
> Please also see our note to novelty in response to reviewer jHas.
>
> ### **Comparison with SELP**
>
> The distinction beyond SELP's LTL-based boolean semantics is discussed in Sec. 5 (L434–437). We further clarify:
>
> - **Different control level:** SELP operates at the symbolic planning level, generating high-level action sequences in natural language. SafeDec enforces safety on *low-level navigation actions* of pretrained embodied policies at every timestep.
> - **Offline vs. online:** SELP requires compiling LTL specs into a Büchi automaton offline. SafeDec performs **real-time inference-time** filtering using STL robustness with no automaton construction or training.
> - **Cost:** SELP incurs 6–8 seconds per plan; SafeDec modifies logits in **milliseconds**, enabling closed-loop execution.
>
> In short, SELP addresses safety **before** execution at the planning level; SafeDec enforces it **during** execution inside transformer policies.
>
> ### **Decoding-time vs. training/control paradigms**
>
> We respectfully disagree that the cited methods address the same problem. SafeDec is the **first method (to our knowledge)** to enforce temporal logic constraints at the **action decoding stage of pretrained transformer policies**, with training-free operation, runtime adaptability to new specifications, and model-agnostic applicability.
>
> - **[2] Meng et. al:** Trains a neural network for MPC with STL objectives. It requires task-specific training, incompatible with pretrained policies without retraining.
> - **[3] Sun et al. (Redriver):** Applies post-hoc gradient-based modifications *after* sampling, disrupting the autoregressive causal chain and creating hidden-state mismatches (as highlighted in Section 3). SafeDec acts *before* sampling in logit space.
> - **[4] Meng et al. (TGPO):** Retrains from scratch with STL-shaped rewards that can involve expensive retraining, inability to adapt to novel specifications at test time, and risk of degrading the  policy's generalization capabilities. SafeDec avoids this exact paradigm.
>
> We will expand related work to make these distinctions explicit.
>
> ---
>
> ## **W2 — Experimental Scope**
>
> The evaluation goes beyond obstacle avoidance and includes **diverse temporal and history-dependent constraints** across four STL specifications (Table 1): **ϕ_avoid** (multi-region safety invariants), **ϕ_ordered** (temporal ordering), **ϕ_no_return** (history-dependent), and **ϕ_geofence** (global state). ϕ_ordered and ϕ_no_return are **non-Markovian** and cannot be reduced to simple obstacle avoidance.
>
> We evaluate across **three state-of-the-art base policies** spanning three training paradigms over **200 procedurally generated environments** per experiment (a new layout per enviroment), covering open-vocabulary object-goal navigation, room-to-room traversal, and attribute-conditioned tasks. We further evaluate on **SafetyChores**  benchmark tasks.
>
> ---
>
> ## **W3 — Significance of RCD vs HCD**
>
> The intended comparison is: **preserving task performance while improving safety vs. the unconstrained policy**. Four one-sided proportion z-tests (n=200 per cell):
>
> | Test | Result |
> |---|---|
> | 1. RCD success > HCD success | 0/9 significant (p<0.05) |
> | 2. HCD STL satisfaction > RCD STL satisfaction | **9/9 significant (p<0.0003)** |
> | 3. HCD success drop vs. unconstrained | **6/9 significant (p<0.05)** |
> | 4. RCD success drop vs. unconstrained | **0/9 significant** |
>
> **HCD** drops task success significantly in 6/9 conditions (e.g., SPOC ϕ_avoid: −10.0 pp, p=0.0083). **RCD** drops success significantly in **0/9 conditions**, and hence does not significantly reduce task success. Although individual RCD-vs-HCD success comparisons are underpowered, RCD exceeds HCD in all 9/9 cases; a one-sided sign test gives **p = 0.002**, supporting a consistent directional advantage. We will add these p-values to Table 2.
>
> ---
>
> ## **Q1 — Failure Modes**
>
> Task success is defined by the navigation benchmark (reaching the goal within the episode horizon), independent of STL satisfaction. Failures arise from: (1) **over-constraining (HCD)**:hard masking can eliminate all forward-progress actions, causing the agent to not find the object before max episode length is reached (2) **infeasible configurations**: randomly sampled safety regions can block all feasible paths to the goal, limiting success regardless of the policy used.
>
> ---
>
> ## **Q2 — Initialization and Violation Regions**
>
> Forbidden regions are placed at random positions with a minimum separation from the start equal to one agent body-width, ensuring the agent does not begin inside a forbidden region. The white dot in the appendix figure is for visualization clarity and is not drawn to scale.
>
> ---
>
> ## **Summary**
>
> We will revise the paper to clarify novelty relative to SELP and related work, and add statistical analysis results to Table 2.

---

> > ### Author Rebuttal · Reviewer_evwF · 2026-04-03
> >
> > Dear authors. Thank you for the diligent rebuttal. I will raise my score to the Weak-acceptance side.
> >
> > There have been debates in the novelty not only from me but also from other reviewer.
> >
> > The authors argued that the millisecond unit, real-time inference seems promising, and there are some advantages compared to [2-4], and I accept these arguments.
> >
> > Why I choose to accept the side:
> > > I think reducing the cost of safety from the simple approach is quite beneficial in robotics and the embodied control problem, though there are still concerns about the rule-making costs. As neuro-symbolic approaches are getting attention especially for the embodied control problem, the low-cost real-time decision making will make beneficial for this area.
> >
> > Though I have no objection if the final decision maker decides to reject the paper
> > > The concerns are still valid with respect to the rule-making costs, and somehow, one may still view it as an incremental approach. In an abstract sense, it is one of the shielding approaches which are prominent in Safe RL literature (e.g., [a-c]). In this sense, I considered a novelty from temporal logic's benefits of low costs, besides of the rule-generating costs.
> >
> > ### **Refs.**
> > [a] Alshiekh, Mohammed, et al. "Safe reinforcement learning via shielding." Proceedings of the AAAI conference on artificial intelligence. Vol. 32. No. 1. 2018.
> >
> > [b] Yang, Wen-Chi, et al. "Safe reinforcement learning via probabilistic logic shields." arXiv preprint arXiv:2303.03226 (2023).
> >
> > [c] Jansen, Nils, et al. "Safe reinforcement learning using probabilistic shields." 31st International Conference on Concurrency Theory (CONCUR 2020). Schloss Dagstuhl–Leibniz-Zentrum für Informatik, 2020.

---

> > > ### Author Response · Authors · 2026-04-08
> > >
> > > We thank the reviewer for their deep engagement with our work and rebuttal. We also deeply appreciate the score raise. We address the three points brought up in the response.
> > >
> > >
> > > Novelty beyond TL benefits of low costs:  **temporal logic is useful not only because it can express safety rules, but also allows a mathematical language to compose them**. In embodied settings, practitioners rarely want to specify just a single forbidden state; they want to combine requirements such as “always avoid unsafe regions,” “stay within an operating zone,” “do not enter region $R_2$ until reaching $R_1$,” etc. TL supports exactly this kind of structured rule composition through Boolean and temporal operators, allowing designers to build richer specifications out of simpler reusable predicates. Second, TL provides a compact language for expressing **history-dependent, sequential, and contextual requirements** over trajectories. Our evaluated specifications go beyond pointwise unsafe-state rejection and require reasoning over execution history and trajectory prefixes. They are also qualitatively different from the reward/cost formulations common in safe MDPs: instead of encoding safety as a scalar penalty or expected constraint cost, TL allows one to specify **logical structure over entire trajectories**, including ordering and conditional dependencies that are awkward to express faithfully with per-step rewards or costs.
> > >
> > > Rule authoring cost concern: We agree this is an important problem, but we do not believe it negates the value of our setting. The key practical question is whether there is value in a method that can consume explicit rules at deployment time without retraining. We believe the answer is yes. Recent work is increasingly reducing authoring burden through automatic generation of formal specifications from high-level instructions [1-2] and our technique provides a complementary systematic approach to enforce those authored specifications.
> > >
> > > Comparison to shielding: While classical Safe RL shielding [a-c] is highly relevant prior work, SafeDec enables additional capabilities from a practical standpoint. The key advantage is **deployment-time adaptation to new specifications without retraining the base policy**. In many safe-RL approaches, safety is encoded through rewards and costs during training, so changing the specification typically requires re-optimization or re-learning to reflect the new constraint. By contrast, SafeDec keeps the pretrained policy fixed and incorporates new temporal-logic rules directly at inference time through decoding. This distinction matters for large pretrained policies, where retraining is expensive, may degrade zero-shot capabilities, and is poorly suited to rapid rule updates.
> > >
> > > If our answers help position the work better, please consider adjusting the score but we also appreciate the existing consideration and engagement from your end!
> > >
> > > - [1] Chen, Yongchao, et al. "Nl2tl: Transforming natural languages to temporal logics using large language models." Proceedings of the 2023 Conference on Empirical Methods in Natural Language Processing. 2023.
> > > - [2] Fang, Yue, et al. "Enhancing transformation from natural language to signal temporal logic using LLMs with diverse external knowledge." Findings of the Association for Computational Linguistics: ACL 2025. 2025.

---

### Decision · Program_Chairs · 2026-04-30

**Decision:**

Accept (regular)

**Comment:**

This paper introduces SafeDec, a constrained decoding framework for transformer-based robot navigation policies that enforces STL safety specifications at inference time without requiring retraining. Rather than performing post-hoc filtering of unsafe actions after sampling, SafeDec regulates the decoding process to ensure generated action sequences align with safety requirements.

The reviewers generally acknowledged the practical merits of SafeDec, highlighting its real-time performance, training-free adaptation, and policy-agnostic nature. The framework's ability to enforce STL specifications on transformer policies without costly fine-tuning was seen as a contribution to the field of embodied AI.

Despite these strengths, several concerns were raised. Multiple reviewers questioned the conceptual novelty, viewing the work as an incremental adaptation of NLP-based constrained decoding to the robotics domain. While the authors argued that grounding logic in physical state space is a non-trivial advancement, some reviewers remained unconvinced. Furthermore, the reliance on an approximate dynamics model and robot-state/localization information was viewed by some reviewers as a practical bottleneck for broader generalization. The lack of physical robot experiments was a noted limitation, as the evaluation was conducted entirely within the AI2-THOR simulation.

Given that three reviewers lean toward Weak Accept (4, 4, 4) and one maintains Reject (2), I recommend Weak Accept.